# Estimation of mean population salt intakes using spot urine samples and associations with body mass index, hypertension, raised blood sugar and hypercholesterolemia: Findings from STEPS Survey 2019, Nepal

**Saroj Bhattarai**[1], **Bihungum Bista**[1], **Binod Kumar Yadav**[2], **Pradip Gynawali**[1], **Anil Poudyal**[1], **Anjani Kumar Jha**[1], **Meghnath Dhimal**[1]*

**1** Nepal Health Research Council, Ramshah Path, Kathmandu, Nepal, **2** Department of Biochemistry, Maharajgunj Medical Campus, Institute of Medicine, Tribhuvan University, Maharajgunj, Kathmandu, Nepal

\* meghdhimal@nhrc.gov.np, meghdhimal@gmail.com

## Abstract

### Background

High dietary salt intake is recognized as a risk factor for several non-communicable diseases (NCDs), in particular cardiovascular diseases (CVDs), including heart attack and stroke. Accurate measurement of population level salt intake is essential for setting targeted goals and plans for salt reduction strategies. We used a spot urine sample to estimate the mean population salt intake in Nepal and evaluated the association of salt intake with excess weight, hypertension, raised blood sugar and hypercholesterolemia, and a number of socio-demographic characteristics.

### Methods

A population-based cross-sectional study was carried out from February to May 2019 using a WHO STEPwise approach to surveillance. Spot urine was collected from 4361 participants aged 15–69 years for the analysis of salt intake. We then used the INTERSALT equation to calculate population salt intake. Student's 't' test, one-way ANOVA and multivariable linear regression were used to assess the association between salt intake and a number of factors. Statistical significance was accepted at $P < .05$.

### Results

The average (±SD) age of participants was 40 (14.1) years. Mean salt intake, derived from spot urine samples, was estimated to be 9.1g/d. A total of 70.8% of the population consumed more than the WHO's recommended amount of 5g salt per day, with almost one third of the population (29%) consuming more than 10g of salt per day. Higher salt intake was significantly associated with male gender (β for male = 0.98g; 95%CI:0.87,1.1) and younger age groups (β$_{25-39 \text{ years}}$ = 0.08; 95%CI:-0.08,0.23) and higher BMI (β = 0.19; 95%

**Data Availability Statement:** The data for this study was obtained from Non Communicable

Diseases Risk Factors: STEPS Survey Nepal 2019.
The data and other additional information about
this study can be received from the Health
Research Section of Nepal Health Research Council
(NHRC) Ramshah Path, Kathmandu, Nepal (email:
research@nhrc.gov.np). The data of this study is
also publicly available from the WHO NCD
Microdata Repository (https://extranet.who.int/
ncdsmicrodata/index.php/catalog/771/data_
dictionary).

**Funding:** The author(s) received no specific
funding for this work.

**Competing interests:** The authors have declared
that no completing interests exist.

CI:0.18,0.21). Participants who were hypertensive and had raised blood cholesterol consumed less salt than people who had normal blood pressure and cholesterol levels (P<0.001).

## Conclusions

Salt consumption in Nepal is high, with a total of 70.8% of the population having a mean salt intake >5g/d, well above the World Health Organization recommendation. High salt intake was found to be associated with sex, age group, education, province, BMI, and raised cholesterol level of participants These findings build a strong case for action to reduce salt consumption in Nepal in order to achieve the global target of 30% reduction in population salt intake by 2025.

## Introduction

Sodium is the principal cation in extracellular fluid in the body and is an essential nutrient necessary for normal cell function and neurotransmission [1]. However, despite being a major source of sodium, high dietary salt intake is associated with high blood pressure, which is recognized as a risk factor for non-communicable diseases (NCDs), in particular cardiovascular diseases (CVDs), including heart attack and stroke [2, 3]. In addition to hypertension, there is also evidence of associations between excessive salt consumption and other CVD risk factors including obesity, diabetes and hypercholesteremia, as well as other health issues such as osteoporosis, cataracts, kidney stones, and gastric cancer, making it a major public health problem [4, 5]. It is concerning therefore, that populations around the world are consuming excessive amounts of salt, with an estimated global mean salt intake of around 9 to 12g/day in several nations [6], and high mean intakes in regions such as Asia, that have been linked to detrimental health impacts [7].

In Nepal, NCDs are estimated to cause 60% of total mortality, with CVDs contributing 22% of all recorded deaths [8]. The population prevalence of NCDs in Nepal has been found to be high, with 11.7% suffering from COPD (11.7%), diabetes mellites (8.5%), chronic kidney disease (6%) and coronary artery diseases (3%) [9]. The 2019 World Health Organization (WHO) STEPwise approach to surveillance of non-communicable disease risk factor (STEPS) survey reported almost 24.5% of the Nepalese population had raised blood pressure [10, 11]. This is concerning, in particular as hypertension and CVDs have been found to place a large economic burden, arising from increasing health-care costs, lowering workplace productivity, increasing sick days, and inflicting permanent disability, on the country [12, 13].

In light of the health burden arising from excess salt intake, the WHO recommends consuming less than 2 grams of sodium, or 5 grams of salt per day, for adults [14]. A 30% relative reduction in mean population intake of salt/sodium by 2025, relative to 2010 levels, under the WHO global action plan, is seen as one of the single most effective public health strategies to reduce the burden of NCDs worldwide [14]. Nepal has incorporated a 5-year multisectoral action plan for prevention and control of NCDs (2014–2020) that includes these salt intake targets [15]. However, to achieve these objectives accurate measurement of salt intake is essential for setting targeted goals and plans for salt reduction strategies [16].

Dietary assessment surveys and nutrition database methods for the estimation of salt intake often underestimate sodium intake for a number of reasons [17]. The standard approach to measuring the mean salt intake of a population has been the collection of 24-h urine samples on a subset of individuals [18]. However, this method is troublesome, time-consuming, costly to participants due to the complex nature of urine sample collection and may miss sodium

excreted through non-urinary routes [19]. In addition some dietary sodium derives from sources other than salt [20]. There has been a growing interest in finding less costly and burdensome alternatives to 24-h urine collection, such as spot urine samples [21]. Equations that use spot urine samples to estimate population salt intake have been explored as a possible alternative in a number of studies [22–25] with the WHO including spot urine as a measure to estimate mean population salt intake, in the STEPS survey protocol in December 2013 [26]. In the present study, we collected spot urine samples from a nationally representative population for the first time in Nepal, in order to estimate mean population salt intake. We also evaluated the associations of salt intake with cardiovascular disease risk factors including excess weight (BMI), hypertension, raised blood sugar and cholesterol levels, as well as their relationships with socio-demographic characteristics.

## Methods

### Study design and sampling technique

A population-based cross-sectional study was carried out from February to May 2019 using a WHO STEPS survey, further details on methodology can be found elsewhere [11]. The survey population includes men and women aged 15–69 years who have been living at their place of residence for at least six months, although visitors who stayed in the households the night before the survey were also eligible for interview. A national representative sample was selected using multistage cluster sampling. The sample size was calculated using the WHO STEPS sample size calculator, such that it was sufficient for analysis at a maximum of 7 strata (7 provinces). The sample size of 925 survey participants from each of seven provinces lead to a total sample size of 6475 participants aged 15 to 69 years. For the sampling process, the ward (the smallest administrative unit) was considered as the primary sampling unit (PSU). All households in the 259 primary sample units (PSUs) were listed using probability proportionate to size (PPS). From this list, 25 households, from 37 PSUs, in each of the seven Provinces were sampled using systematic random sampling. From each of the selected households, one eligible participant (15–69 years) was chosen randomly and approached for the study.

### Data collection

The survey was conducted using the standardized WHO NCD STEPS questionnaire version 3.2 and a team of 60 field research assistants with health sciences backgrounds. All field research assistants received training in conducting all parts of the STEPS data collection procedure from WHO and Nepal Health Research Council (NHRC) technical expert team. The data collection process included three STEPS: STEP 1 comprised face to face interview to complete a questionnaire on demographic and behavioral characteristics of the study population. STEP 2 involved the collection of anthropometric measurements including blood pressure, height, weight, hip, and waist circumference. Height and weight were measured with a portable digital weighing scale (Seca, Germany). Waist and hip circumference were measured using a constant tension tape (Seca, Germany). Whilst blood pressure was measured using a digital, automated blood pressure monitor (OMRON digital device) with a universal size cuff. STEP 3 included biochemical measurements for diabetes, raised blood glucose and cholesterol level. Blood glucose and total cholesterol were measured through dry chemistry using CardioCheck PA Analyser, concentrations of glucose and total cholesterol were measured in capillary whole blood, as recommended and supported by the WHO. Blood samples were collected at STEP 3 the day after completion of STEPS 1 and 2, with participants instructed to fast overnight for 12 hours.

### Estimation of 24-hour salt intake based on spot urine testing

For the estimation of the mean population salt intake, the STEPS survey utilized a spot urine sample as a proxy to 24-hour urine samples. Spot urine collection was done to identify the level of Sodium (Na), potassium (K), and creatinine.

**Process.** For urine collection, a 10 ml urine container with a QR code (Identification number) was provided to participants to collect spot urine after they had completed STEPS 1 and 2 of the data collection. Participants self-collected the urine samples at home, before fasting for the blood sample collection, and were then asked to bring the urine sample the following day to their scheduled appointment for collection of the biochemical measurements.

The collected urine samples were stored in a dark place at normal room temperature until they were transported to the laboratory that had been set up in every province headquarters to analyse urine collected in that province. Determination of Na and K in the urine was carried with Ion-selective Electrodes (ISE) in an automated Analyzer (Beckman Coulter, CA, USA). Levels of creatinine in the urine were estimated using a semi-automated biochemistry analyzer (Nova Biomedical Cooperation, Waltham, MA, USA). The unit of measurements for Na and K was mmol/L, for creatinine it was mg/dl. Participants who were pregnant, those who fasted before collecting the urine sample, and those who samples were contaminated with blood were excluded at the time of analysis. Participants whose height was less than 100 cm or greater than 270 cm, and those whose weight was less than 20kg or above 350 kg were also excluded.

### Daily salt consumption estimation

This is the first time that 24-hour salt intake has been estimated in Nepal, using spot urine samples. Kawasaki, INTERSALT, and Tanaka are the three key studies that set up the estimation of 24-hour urinary sodium intake from spot urine samples and that provide equations for 24 hour sodium intake estimation [22, 24, 25]. However, limited evidence supports the preferential use of one equation over another in a given population/context [27]. For this survey, we used the INTERSALT Southern European equation to estimate 24 hours mean salt intake because it was developed using a large heterogeneous population sample [22]. In addition, the same equation had been used in previous survey rounds, thereby facilitating comparison of results and assessment of trends.

### INTERSALT Southern European equation

The equations given below are used to with the spot urine test to compute 24-hour 'sodium' intake, this is then converted to 'salt' intake by the division of 17.1 (or multiplication of 2.54/1000*23) as a conversion factor to obtain the final estimated 24-hour salt intake in grams.

**Male.**

$$\left(20.861 + 0.45 \times Naspot\left(\frac{mmol}{L}\right)\right) - 3.09 \times Crspot\left(\frac{mmol}{L}\right) + 4.16 \times BMI\left(\frac{kg}{m^2}\right) + 0.22$$
$$\times Age(year)$$

**Female.**

$$\left(21.98 + 0.33 \times Naspot\left(\frac{mmol}{L}\right)\right) - 2.44 \times Crspot\left(\frac{mmol}{L}\right) + 2.42 \times BMI\left(\frac{kg}{m^2}\right) + 2.34$$
$$\times Age(year) - 0.03 \times Age^2(year)$$

*Naspot*: Sodium concentration in spot urine (mmol/L)
*Crspot*: Creatinine concentration in spot urine (mmol/L)

**BMI:** Body Mass Index

## Data processing and analysis

Descriptive statistics for demographics and collected data were recorded for all participants who consented to STEP 1 of the survey. All the analyses for this study were performed on STATA 13.1 version using a survey (svy) set command, defining clusters, and sampling weight information. The proportion of the population above the WHO-recommended guideline of 5 g/d of salt and sodium to potassium ratio were measured.

Student's 't' test, one-way ANOVA, and Chi-square statistics were used to determine the association of mean salt consumption levels with explanatory factors. The relationship between salt intake and participant characteristics was explored using multivariable linear regression, these characteristics included age category, sex, province, education, income, BMI, blood pressure, raised blood sugar, and cholesterol level. Statistical significance was accepted at $P < .05$.

## Ethical consideration

Written informed consent was obtained from each of the research participants jointly for STEPS 1 and 2 and then separately for STEPS 3. Participants were informed regarding their right to withdraw from the study at any time without penalty and that confidentiality and consent will be upheld in accordance with ethical research standards. Ethical approval of original survey was obtained from the Ethical Review Board (ERB) of NHRC, Government of Nepal (Registration number 293/2018).

## Results

### Response rate

Amongst the initially planned 6475 sample size, 1 PSU with 25 participants was dropped due to heavy snow, leaving 6450 as our total sample size. The number of participants who consented and completed the survey for STEP 1 was 5593 (86.7%), for Step 2 was 5582 (86.5%), and STEP 3 was 5350 (82.6%). Among them, only 4361 (67.6%) participants consented to provide the spot urine for analysis of salt intake. The demographic characteristics of the participants included in this study are displayed in "Table 1".

Characteristics of the 1998 male and 3595 female participants in the study are reported in "Table 1". The average (±SD) age of participants was 40 (14.1) years. Most of the participants (both male and female) reside in a municipality and had none/primary education. The mean BMI of females is slightly greater than that of males (22.8 kg/m$^2$ compared to 22.6 kg/m$^2$), in contrast, mean waist to hip ratio was slightly higher in males (0.92) compared to females (0.89). Mean blood pressure (BP) was higher in males than females (127.7/83 mm of Hg compared to 121.3/80.3 mm of Hg). Among the male respondents, more than one-fourth (29.8%) of the population was assessed as having hypertension (i.e., SPB $\geq$140 mm Hg and/or DBP $\geq$90 mm Hg). Similarly, among the female respondents, about 19.7% participants had high blood pressure. More female participants (11.2%) reported having previously been diagnosed with hypertension and to be receiving treatment compared to (7.0%) male participants. Raised blood sugar was more prevalent among males than females (6.3% compared to 5.3%) in contrast; a greater proportion of female than male participants had raised blood cholesterol levels (13.9% compared to 7.7%).

Mean salt intake was estimated to be 9.1g/d derived from spot urine samples (Table 2). Males had significantly higher salt intake than females (9.6 g/d compared to 8.7 g/d). A total of 70.8% of the population consumed more than the WHO's recommended amount of 5g salt

**Table 1. Study population characteristics.**

| Characteristics | Male | N | Female | n |
|---|---|---|---|---|
| | % | | % | |
| **Socio-demographic Information (STEPS 1)** | | | | |
| **Age*** (years) | 42.2 (14.6) | 1998 | 38.6 (13.7) | 3595 |
| **Age range (years)** | | | | |
| 15–24 | 13.8 | 275 | 15.8 | 568 |
| 25–39 | 30.8 | 615 | 41.0 | 1,472 |
| 40–54 | 30.5 | 609 | 26.8 | 965 |
| 55–69 | 25.0 | 499 | 16.4 | 590 |
| **Residence** | | | | |
| Metropolitan/Sub metropolitan city | 13.8 | 276 | 11.9 | 429 |
| Municipality | 48.3 | 964 | 49.8 | 1,791 |
| Rural municipality | 37.9 | 758 | 38.3 | 1,375 |
| **Province** | | | | |
| Province 1 | 14.3 | 285 | 14.4 | 519 |
| Province 2 | **17.7** | 353 | 12.5 | 450 |
| Bagmati Province | 15.1 | 302 | 12.7 | 457 |
| Gandaki Province | 13.4 | 267 | 14.6 | 526 |
| Lumbini Province | 13.4 | 268 | 14.7 | 529 |
| Karnali Province | 13.1 | 261 | 15.2 | 547 |
| Sudurpashchim Province | 13.1 | 262 | 15.8 | 567 |
| **Education** | | | | |
| None/less than primary | 39.6 | 792 | 55.6 | 2,000 |
| Primary | 21.2 | 424 | 17.4 | 627 |
| Secondary | 23.3 | 466 | 17.3 | 622 |
| More than secondary | 15.8 | 316 | 9.6 | 345 |
| **Wealth** | | | | |
| Lowest | 25.2 | 504 | 32.0 | 1,149 |
| Second | 18.3 | 366 | 19.4 | 696 |
| Middle | 17.3 | 345 | 16.8 | 604 |
| Fourth | 16.9 | 338 | 15.0 | 540 |
| Highest | 22.3 | 445 | 16.9 | 606 |
| **Physical Measurements (STEPS 2)** | | | | |
| Height* (cm) | 161.1 (7.8) | 1997 | 151.8 (6.6) | 3522 |
| Weight* (kg) | 58.7 (11.1) | 1997 | 52.6 (10.9) | 3522 |
| Waist circumferences* (cm) | 80.4 (11) | 1997 | 79.9 (17.9) | 3522 |
| Waist-hip ratio* | 0.92 (8.5) | 1997 | 0.89 (9.7) | 3521 |
| BMI* (kg/m$^2$) | 22.6 (3.8) | 1992 | 22.8 (4.2) | 3507 |
| Normal (BMI 18.5–24.9) | 65.9 | 1992 | 65.1 | 3507 |
| Underweight (BMI< = 18.4) | 10.7 | 1992 | 9.8 | 3507 |
| Overweight (BMI 25.0–29.9) | 20.2 | 1992 | 19.8 | 3507 |
| Obese (BMI > = 30.0) | 3.2 | 1992 | 5.3 | 3507 |
| Systolic BP* (mm Hg) | 127.7 (17.9) | 1997 | 121.3 (17.3) | 3585 |
| Diastolic BP* (mm Hg) | 83.0 (11.6) | 1997 | 80.3 (10.5) | 3585 |
| Hypertension | 29.8 | 1966 | 19.7 | 3540 |
| People measured to be hypertensive on treatment | 7.9 | 721 | 11.2 | 817 |
| **Biochemical measurements (STEPS 3)** | | | | |
| Raised blood sugar (mg/dl) | 6.3 | 1834 | 5.3 | 3357 |

*(Continued)*

**Table 1.** (Continued)

| Characteristics | Male | N | Female | n |
|---|---|---|---|---|
| | % | | % | |
| Hypercholesterolemia (mg/dl) | 7.7 | 1904 | 13.9 | 3438 |

* Values are shown as mean and standard deviation, mean (± SD).

BP, Blood pressure.

per day, with almost one-third of the population (29%) consuming more than 10g of salt per day. The sodium/potassium ratio for the population was 4.1 (SD, 5.7).

Table 3 shows an analysis of the level of salt intake in relation to participants' characteristics. Salt intake was found to differ significantly ($p < 0.05$) between sex, age range, education, province, BMI, and raised blood cholesterol level of participants. However; there was no significant associations between salt consumption and hypertension, raised blood sugar, and wealth index of participants.

Male participants consumed on average 1g more salt per day than females (Table 4). Young adults (25–39 years) ate 0.08g more salt per day than those in the 55–69 years age group ($P < 0.001$). Participants with a higher than secondary education ate less salt compared to participants who had a primary or secondary education as their highest level. Similarly, salt intake decreased as the wealth index of the participant's increased (Table 4). Differences were found between provinces, with participants living in the Bagmati Province having the lowest intake and those in Gandaki Province also having a significantly lower intake than participants in Province 1 (the reference category). Participants living in Karnali and Sudurpashchim provinces had significantly higher intakes than those in Province 1, with Karnali province having the highest mean intakes. Consumption of salt intake increased by 0.2g with every 1 kg/m$^2$ increase in BMI ($P < 0.001$). Participants who were hypertensive and those with raised blood cholesterol ate less salt than people who had normal blood pressure and cholesterol levels ($P < 0.001$). Compared to participants with normal blood sugar levels, those with raised blood sugar ate less salt, although this difference was not found to be significant (Table 4).

## Discussion

To the best of the authors' knowledge, this is the first nationwide population study to estimate mean population-level salt intake using spot urine samples in the Nepalese population. Estimation of salt intake based upon spot urine samples elsewhere, have shown excellent concordance

**Table 2.** Weighted results for salt intake, potassium intake, and sodium/potassium ratio.

| Estimated salt intake (g/day) | Overall (4361) | Male (1600) | Female (2761) |
|---|---|---|---|
| | mean ± (SD) or % | mean ± (SD) or % | mean ± (SD) or % |
| | 9.1(1.8) | 9.6(2.0) | 8.8(1.6) |
| Salt intake above the 5g WHO target (g/day) | 70.8% | 59.6% | 77.2% |
| Salt intake above the 10 g WHO target (g/day) | 29% | 40% | 22.7% |
| Potassium intake, mmol | 48.9(32.6) | 52.1(36.2) | 46.0(30.3) |
| Sodium intake, mmol | 120.8(80.5) | 111.6(74.4) | 126(83.3) |
| Sodium/potassium ratio, | 4.1 (5.7) | 3.6 (5.2) | 4.3 (5.9) |
| Creatinine, mmol | 3.9 (3.7) | 4 (4.3) | 3.8 (3.3) |

S.D, Standard deviation.

**Table 3. Association between levels of salt intake with participants' characteristics.**

| Group | Overall (n) | Mean, 95%-CI | P value |
|---|---|---|---|
| **Sex** | | | |
| Women | 2761 | 8.8 (8.7–8.9) | <0.001 |
| Men | 1600 | 9.6 (9.5–9.7) | |
| **Age range (years)** | | | |
| 15–24 | 614 | 9.0 (8.8–9.1) | <0.001 |
| 25–39 | 1617 | 9.3 (9.2–9.4) | |
| 40–54 | 1245 | 9.2 (9.1–9.3) | |
| 55–69 | 885 | 8.7 (8.5–8.8) | |
| **Education** | | | |
| None/less than primary | 2152 | 8.9 (8.9–9.0) | <0.001 |
| Primary | 829 | 9.2 (9.1–9.4) | |
| Secondary | 858 | 9.3 (9.2–9.4) | |
| More than secondary | 522 | 9.2 (9.1–9.4) | |
| **Wealth index** | | | |
| Lowest | 1281 | 9.1 (9.0–9.2) | 0.257 |
| Second | 831 | 9.2 (9.1–9.3) | |
| Middle | 749 | 9.0 (8.9–9.2) | |
| Fourth | 670 | 9.0 (8.9–9.2) | |
| Highest | 830 | 9.1 (9.0–9.2) | |
| **Province** | | | |
| Province 1 | 711 | 9.1 (8.9–9.2) | <0.001 |
| Province 2 | 713 | 8.9 (8.8–9.0) | |
| Bagmati Province | 674 | 9.0 (8.9–9.1) | |
| Gandaki Province | 726 | 9.1 (8.9–9.2) | |
| Lumbini Province | 96 | 8.7 (8.4–9.1) | |
| Karnali Province | 717 | 9.4 (9.3–9.6) | |
| Sudurpashchim Province | 724 | 9.1 (9.0–9.2) | |
| **BMI (kg/m$^2$)** | | | |
| Underweight | 372 | 8.0 (7.8–8.2) | <0.001 |
| Normal | 2771 | 8.9 (8.8–8.9) | |
| Overweight | 962 | 9.7 (9.6–9.8) | |
| Obese | 240 | 10.6 (10.4–10.9) | |
| **Hypertension** | | | |
| No | 3086 | 9.1 (9.0–9.2) | 0.735 |
| Yes | 1220 | 9.1 (9.0–9.2) | |
| Raised blood sugar | | | |
| No | 3917 | 9.1 (9.1–9.2) | 0.533 |
| Yes | 255 | 9.0 (8.8–9.3) | |
| Hypercholesterolemia | | | |
| No | 3770 | 9.2 (9.1–9.2) | <0.001 |
| Yes | 539 | 8.8 (8.6–8.9) | |

with 24-hr urine collection measurements [28] and had outstanding sensitivity (97%) and specificity (100%) at classifying mean population salt intake as above or below the WHO maximum guideline value of 5 g/day [20]. Our findings demonstrated the substantial heterogeneity in average population level salt intake in the Nepalese population. We found mean salt intake to be 9.6g/day for males and 8.7g/day for females, with the mean salt intake of 9.1g/day among

**Table 4. Multivariable analysis of salt intake with participants' characteristics.**

| Characteristics | Coefficient (β) | S.E | P-Value | 95%-CI |
|---|---|---|---|---|
| **Sex** | | | | |
| Female | Reference | | | |
| Male | 0.98 | 0.05 | <0.001 | (0.87–1.1) |
| **Age range (years)** | | | | |
| 15–24 | Reference | | | |
| 25–39 | 0.08 | 0.08 | 0.333 | (-0.08–0.23) |
| 40–54 | -0.03 | 0.09 | 0.737 | (-0.20–0.14) |
| 55–69 | -0.43 | 0.10 | <0.001 | (-0.63- -0.24) |
| **Education** | | | | |
| None/less than primary | Reference | | | |
| Primary | 0.01 | 0.07 | 0.846 | (-0.12–0.15) |
| Secondary | 0.23 | 0.08 | 0.756 | (-0.12–0.17) |
| More than secondary | -0.11 | 0.09 | 0.206 | (-0.29–0.06) |
| **Wealth index** | | | | |
| Lowest | Reference | | | |
| Second | 0.09 | 0.07 | 0.205 | (-0.05–0.24) |
| Middle | -0.07 | 0.08 | 0.389 | (-0.22–0.09) |
| Fourth | -0.19 | 0.08 | 0.024 | (-0.35- -0.02) |
| Highest | -0.31 | 0.09 | <0.001 | (-0.48- -0.14) |
| **Province** | | | | |
| Province 1 | Reference | | | |
| Province 2 | -0.05 | 0.09 | 0.543 | (-0.22–0.12) |
| Bagmati Province | -0.31 | 0.09 | <0.001 | (-0.49- -0.14) |
| Gandaki Province | -0.20 | 0.08 | 0.019 | (-0.3- -0.03) |
| Lumbini Province | -0.25 | 0.17 | 0.153 | (-0.59–0.09) |
| Karnali province | 0.52 | 0.09 | <0.001 | (0.34–0.69) |
| Sudurpashchim Province | 0.28 | 0.09 | <0.001 | (0.11–0.45) |
| **BMI** | 0.19 | 0.01 | <0.001 | (0.18–0.21) |
| **Hypertension (No)** | Reference | | | |
| **Hypertension (Yes)** | -0.28 | 0.06 | <0.001 | (-0.40- -0.16) |
| **Raised blood sugar (No)** | Reference | | | |
| **Raised blood sugar (Yes)** | -0.09 | 0.11 | 0.405 | (-0.29–0.12) |
| **Hypercholesterolemia (No)** | Reference | | | |
| **Hypercholesterolemia (Yes)** | -0.32 | 0.08 | <0.001 | (-0.47- -0.17) |
| Constant | 4.53 | 0.18 | <0.001 | (4.12–4.87) |

S.E, Standard Error.

adult Nepalese population, higher than the WHO recommended amount of 5g/day. The majority of the population (70.8%) in our sample consumed more salt than this recommendation, with 29% consuming more than 10g/day, double the WHO recommended value. The average mean salt consumption level in our study is comparable to small-scale studies conducted in Nepal [7], in an Urban South Indian Population [29], in some states of India [30], Bhutan [31], and Korea [32]. The salt intake value obtained in this study is slightly lower than the previous finding reported from Nepal [33], Bangladesh [34], and China [35] using 24-hour urine collection measurement methods. However, both methods of measurements show salt consumption in Nepal is well above the WHO global recommendation, indicating urgent

action is needed to tackle the non-communicable disease crisis in the country and to reduce population-wide dietary salt intake, in order to decrease the number of deaths from hypertension, cardiovascular disease, and stroke. In low- and middle-income countries (LMICs) the major contributing factors to the high amount of daily salt intake comes from discretionary salt used during cooking and salting food at the table [36]. A study from south India demonstrated that food items; pulse-based dishes, cereal-based dishes, and vegetable-based dishes are the major contributors to daily salt intake [37]. It is likely that similar sources may be key in the high salt intake in the Nepalese population. It has already been reported that intake of low sodium salt is beneficial to health in many aspects [38]. Thus, it is recommended to create awareness among the general public in LMICs to cut down the use of discretionary salt in foods to decrease the level of salt intake [36, 39].

Our finding reported the sodium/potassium (Na/K) ratio for the population studied was 3.4 (SE, 0.2), which is consistent with the finding in a previous study done by Samoa et al. [40]. Reducing the Na/K ratio is essential for preventing hypertension and cardiovascular disease; however, there is no generally accepted recommended guideline for the Na/K ratio [41]. Future studies are required to establish recommended levels for the Na/K ratio, to inform individuals regarding the risk of hypertension and cardiovascular disease [42]. The finding of this study suggesting a higher salt intake in males than females is consistent with other studies [7, 32, 33, 43]. Most of the male population in Nepal is engaged in outdoor activity and most often consume prepared or ready-to-eat foods, which may lead to greater sodium consumption. Similarly, this study shows that the prevalence of salt intake >5g was higher in females than in males, whereas that of salt intake >10 g was higher in males than in females. These findings might be linked to different lifestyle and behavioral choices between the male and female populations, including women being less likely to exhibit extremes in selection of both type and amounts of food. This study shows young adults (25–39 years) ate more salt as compared to the 15–24 years age group, however, salt consumption declined in the middle (40–54 years) and older (55–69 years) age group participants. There is no previous study available in Nepal to support these results, however it is possible that middle and older age individuals are more likely to experience one or more NCDs in their life time and are likely to be aware of salt intake, they may therefore practice and maintain healthier dietary practices. Similarly, the Nepalese economic structure has changed, shifting away from the agricultural food supply system towards the modern processing food supply system. Trade liberalization has made processed foods easily available at supermarkets and fast food outlets [44]. A study conducted in Lifestyle Practices and Obesity in Nepalese Youth showed that the majority (75.78%) of participants consume fast-food [45]. Similarly, people with higher grade education levels were significantly more likely to be knowledgeable about risk factors of non-communicable disease [46], which supports the low salt intake in participants with a higher education found in this study. Risk factors of NCDs increased with increasing wealth [47, 48], however; in contrast to these findings, our study shows salt consumption declined in those people who were included in the fourth and highest wealth index. This may be due to fact that the wealthier populations may be aware of risk factors of NCD and may change their lifestyle. Our study showed people who reside in Karnali and Sudhurpashchim province ate more salt than other provinces. There is no previous research in Nepal to support these results; one of the possible reasons could be different dietary patterns and dietary habits of people by province. This should be a proposed area of future research, as regional and provincial differences would impact the success of any population level interventions. Salt intake increased as BMI increased in line with previous studies, demonstrating that salt intake was higher in overweight and obese individuals, this may lead to susceptibility to NCD in later life among those population [30, 33, 49]. The correlation of high salt intake with obesity is well known, but the biological mechanisms behind this

correlation are not well understood yet. One of the possible reasons for increased sodium intake as BMI increased in this study might be attributed to unhealthy lifestyles, including poor diet, physical inactivity, and sedentary behaviors of the study population [50].

Unlike other studies conducted elsewhere [51, 52], we didn't find any significant association between increased salt intake and hypertension. This may be due to the fact that hypertensive individuals are aware of the amount of salt intake as they are advised by the treating physician and dietician about the amount of salt intake in their diet. Patients with raised blood pressure may be under medical advice to restrict sodium intake, or that phenotype of salt sensitivity is heterogeneous, influenced from genetic to environmental factors with multiple mechanisms that potentially link high salt intake to increases in blood pressure [53]. However, our finding reported salt intake was less among the hypertensive population compared to non-hypertensive. This may be due to reverse causation but as our study is cross-sectional, it is not possible to test this hypothesis. Sodium reduction has been found to substantially lower blood pressure, even among those with starting systolic blood pressure levels as low as 120 mm Hg [54]. These findings indicate potentially important health benefits from sodium reduction among normotensive as well as hypertensive individuals [54]. More importantly, sodium reduction among normotensive individuals could potentially avert or delay the development of hypertension with aging, as the association between sodium intake and blood pressure is greater at older age [55]. High-salt intake is a major risk factor for developing hypertension in type 2 diabetes mellitus, but its effects on glucose homeostasis are controversial [56]. Similarly, we did not find an association of salt intake with the raised blood sugar in the population though there was a lower salt intake in those with raised blood sugar compared to those with normal blood sugar levels. This study found an association between salt intake and hypercholesterolemia, however, there is no established conclusive evidence regarding the relationship between sodium intake and hypercholesterolemia [57, 58]. Studies have found mixed evidence between reducing sodium intake and hypercholesterolemia as well as in associations between sodium intake and all-cause mortality, incidence of cardiovascular disease and non-fatal coronary heart disease [58, 59].

This study has several strengths, first it is a nationwide population study, that incorporated individuals of all provinces, caste, sex, and age, and thus the study findings are highly generalizable for Nepal. Second, to the best of our knowledge, this is the first study in the Nepalese population to focus on the association of salt intake with different characteristic. This is an important step in determining future interventions to reduce salt intake among the Nepalese population. However, this study also has a number of limitations; it is a cross-sectional study, so cannot attribute the causality from the association of estimated salt intake with BMI and blood pressure. Second, we used spot urine sample for the estimation of salt intake instead of the gold standard method 24-h urine sample. Many large epidemiological studies have already adopted the estimation of salt intake using spot urine samples, instead of 24-h urine samples, because of the ease of urine sample collection and participant enrollment. Third, this study has included hypertensive individual but their detailed information on the antihypertensive drugs was not inclusively collected. It is postulated that some antihypertensive drugs with natriuretic properties could be impacted urinary sodium levels, whereas other antihypertensive drugs may not. This should be explored further in future research. Fourth, we have not focused on the status of renal function tests (RFT) of the participants, which might have an influence on the estimation of salt intake.

WHO is considering the use of spot urine samples to assess population salt intake as part of STEPS, in order to monitor progress toward the global targets. Several equations have been developed and tested and proven potentially useful for estimating mean population 24 hour salt intake from spot urine samples [22–25]. However, they have yet to be tested and evaluated

in the Nepalese population. The estimation of salt intake using spot urine samples in this study, indicates that salt intakes are well above the WHO recommendations and may provide a benchmark to assess the impact of salt reduction efforts in Nepal [20]. This study has important clinical and public health implications. In Nepal, almost 24.5% of the Nepalese population had raised blood pressure, and more than 22% of NCDs deaths are related to CVDs [8]. There is strong and consistent evidence that excess salt intake remains one of the key risk factors for high BP and thereby increased risks of CVDs, and stroke [3]. Effective interventions to reduce salt intake at the population level present an opportunity to combat the increasing burden of NCDs and increase life expectancy.

## Conclusions

According to our findings, Nepal has a daily salt intake that is around double that of the WHO's recommended limit. A total of 70.8% of the population had a salt intake >5 g/d. The findings of this study will aid policymakers and other relevant stakeholders in Nepal at the federal, provincial and local levels in tracking trends and progress toward salt intake targets, as well as guiding future policies and initiatives aimed at reducing salt intake among the population. Furthermore, these findings build a strong case of action needed to reduce salt consumption in Nepal to achieve the global target of a 30% reduction in population salt intake by 2025.

## Acknowledgments

We would like to acknowledge the effort of all the individuals involved in this survey, express deep sense of appreciation to the steering committee and technical working group (TWG) members. We would like to express sincere thanks to Dr. Manju Rani, Regional Advisor (Non-communicable diseases policy, governance and surveillance), Naveen Agarwal (Surveillance Management Associate at WHO/SEARO); Dr. Patricia Rarau (WHO HQ) (training of field enumerators); Dr. Stefan Savin (WHO HQ) (data analysis); Dr. Md. Khurshid Alam Hyder (WHO Nepal) and Dr. Lonim Prasai Dixit (WHO Nepal), Ms. Yvonne Y. Xu, Ms. Preetika D. Banerjee, Ms. Surabhi Chaturvedi, from WHO SEARO for their valuable and remarkable contribution from beginning to the end of this survey. We are also grateful to Dr Nick Townsend from the United Kingdom for proof reading of our manuscript. We acknowledge the support of Government of Nepal and WHO to conduct the survey. Lastly, we would like to thank all NHRC staff that helped throughout this study.

## Author Contributions

**Conceptualization:** Saroj Bhattarai, Pradip Gynawali, Anjani Kumar Jha, Meghnath Dhimal.

**Formal analysis:** Bihungum Bista.

**Methodology:** Anjani Kumar Jha.

**Project administration:** Anjani Kumar Jha, Meghnath Dhimal.

**Supervision:** Meghnath Dhimal.

**Visualization:** Binod Kumar Yadav.

**Writing – original draft:** Saroj Bhattarai.

**Writing – review & editing:** Saroj Bhattarai, Bihungum Bista, Binod Kumar Yadav, Pradip Gynawali, Anil Poudyal, Anjani Kumar Jha, Meghnath Dhimal.

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
