## [Decision Letter · Decision Letter 0]

23 Apr 2021

PONE-D-21-06676

Estimation of mean Population salt intakes using Spot Urine samples and its association to BMI, Hypertension, raised Blood Sugar and Blood Lipids: Findings from Non-communicable Disease Risk Factors (STEPS) Survey 2019 in Nepal

PLOS ONE

Dear Dr. Dhimal,

Thank you for submitting your manuscript to PLOS ONE. After careful consideration, we feel that it has merit but does not fully meet PLOS ONE’s publication criteria as it currently stands. Therefore, we invite you to submit a revised version of the manuscript that addresses the points raised during the review process.

We look forward to receiving your revised manuscript.

Kind regards,

Shyam Sundar Budhathoki, MBBS, MD, MPH

Academic Editor

PLOS ONE

Journal Requirements:

3. Thank you for submitting the above manuscript to PLOS ONE. During our internal evaluation of the manuscript, we found significant text overlap between your submission and the following previously published works, some of which you are an author.

- https://www.bmj.com/content/bmj/368/bmj.m315.full.pdf

- https://onlinelibrary.wiley.com/doi/full/10.1111/jch.12778

Please revise the manuscript to rephrase the duplicated text, cite your sources, and provide details as to how the current manuscript advances on previous work. Please note that further consideration is dependent on the submission of a manuscript that addresses these concerns about the overlap in text with published work.

Reviewers' comments:

Reviewer's Responses to Questions

**Comments to the Author**

1. Is the manuscript technically sound, and do the data support the conclusions?

Reviewer #1: Yes

Reviewer #2: Partly

2. Has the statistical analysis been performed appropriately and rigorously? 

Reviewer #1: No

Reviewer #2: I Don't Know

3. Have the authors made all data underlying the findings in their manuscript fully available?

Reviewer #1: No

Reviewer #2: Yes

4. Is the manuscript presented in an intelligible fashion and written in standard English?

Reviewer #1: Yes

Reviewer #2: Yes

5. Review Comments to the Author

Reviewer #1: This study aims to determine the population mean of salt intakes and its correlates using 2019 STEPS survey data in Nepal. The research question is adequately motivated. However, there are some concerns that authors need to address to improve the quality of the manuscript.

Comments

• The title and objectives of the study are not consistent.

• Title needs some improvement. For example, while hypertension and raised blood sugar indicate a direction, the authors did not mention about BMI and lipid level similarly. Please revise the title. Also, use hypercholesterolemia instead of high cholesterol level.

• It is highly preferable to use consistent terms throughout the whole manuscript. For instance, raised blood sugar is used in the title but there is no indication of such variables in the result, except diabetes.

• The authors mentioned that the equation from INTERSALT was developed using a large heterogenous population sample. Do the authors mean by the term “heterogenous” that the equation can be applied to the Nepalese population?

• The authors also explicitly mentioned the equation INTERSALT for southern Europe. If this is the case, there would be at least some anthropometric differences between these and Nepalese population? For example, considering BMI and age adjustment to the equation would seem to be more appropriate instead of using the original equation.

• In Table 2, the prevalence of salt intake > 5 g was higher in women than in man, whereas that of salt intake > 10 g was reversed, higher in man than in women. Please discuss this.

• As the authors rightly discussed that people with history of hypertension may be aware of their hypertensive status and likely to reduce salt intake. This can also be linked to behavioral and lifestyle choices. That’s why salt intake is higher in men as women are more likely to take care of themselves. Likewise, BMI can also be attributed by lifestyle choices and hence, by salt intake. Therefore, there is no clear indication of exposure and outcome pathway in this model.

• What are the clinical implications of the study beyond the assessment of population salt intake and validation of equation in Nepalese population?

• Conclusion should be based on the study findings. For instance, the study’s results did not present the lifestyle modification and its importance on health indicators. It is also not a finding of the study that “population-based salt reduction strategies are cost-effective and cost-saving in most of the settings for prevention of non-communicable diseases.”. Please revise the conclusion.

• References should be relevant and up-to-date. Please try to reduce the number of references.

• in Table 4, there is no need to report R squared or adjusted R squared without presenting all models. Remove the last two columns.

• If p-value is 0.000, indicate using “<0.001” in Table 3 and 4.

Reviewer #2: General Comments:

This study used data collected from spot urine tests to estimate population-level salt intake in Nepal, and to assess the relationship of salt intake with various anthropometric and metabolic factors. Data was derived from the well-known STEPS Survey and incorporated a large, nationally-representative study population. The study has potentially very relevant and important findings for future intervention for NCDs in Nepal. However, I have a number of concerns and/or queries with some of the information being presented in its current form which I would like the authors to address.

Firstly, in the introduction, there needs to be more emphasis as to why salt intake corresponds to various anthropometric and metabolic variables, so as to build a strong rationale for the investigation into these relationships in this study. Particular attention should be paid to the variables that are subsequently stated in the study aim.

Greater clarity is needed over the statistical methods employed in this study, as at times important data appear to have conflicting findings depending on the analyses used. See specific comments in results section.

Tables need revising for better clarity on the reporting of data, particularly concerning SI units.

The discussion covered the major points raised in the study. However, clear assessment of this section is hampered at this time by outstanding questions over the presentation (and conflicts) of results using ANOVA and multivariable/multivariate regression. Important limitations were highlighted in the discussion, which was good to see, as well as the study strengths. I feel you could make the strengths of this study seem more impactful though, by concentrating less on it being the “first” study to show these findings and more on the relevance, i.e., that it is a Nationwide population study, that incorporated individuals of all regions, caste, sex and age, and thus the study findings are highly generalisable for Nepal. This is an important step in determining future interventions to reduce salt intake among Nepalese peoples.

Although I commend the authors on a generally well-written manuscript, there were several grammatical errors noted throughout and, particularly as PLOS One do not copy edit, the authors should ensure that the written language is reviewed and modified accordingly.

Specific Comments:

Introduction:

Line 87: You state equations have been investigated in several studies but provide no indication of the validity or reliability of these methods. Please expand further by discussing the strengths, limitations and validity and reliability of using estimation methods.

Line 88-89: The grammar isn’t quite right here – I’m not sure what you’re meaning.

Line 91-: You need to be clearer on your aim/purpose. Use future tense (what you aim to do) rather than past tense (what you have already done).

Methods:

Line 102: Was the sample of 925 per province selected in advance, i.e., was this a target? If so, why was this particular number selected? Was there a margin, or mechanism in place in case you weren’t able to recruit exactly this number (i.e., it seems as though only this precise number of households were contacted, but what if they refused to participate in the survey)?

Line 104: Could you provide further clarity on what constitutes a primary sample unit. How were these identified?

Line 107: Please clarify what you mean when you say ‘using the android tablet’. Is this specifically in relation to randomly sampling one adult in each household, or something else? In what way was the android tablet used?

Line 110 onwards: If relevant, please clarify that the field research assistants led/facilitated all or which parts of the data collection (i.e., steps 1-3?). Please also clarify the stated background experience, i.e., presumably, researchers were required to have one of the stated backgrounds rather than expecting to have experience in all of these fields?

Line 113: Further detail is needed on the interview process. How was this conducted? Did the researchers ask questions and complete the answers for the participants, or were they just ‘on-hand’ to answer any questions if they arose whilst the participants completed the questionnaires? Were consistent methods applied throughout Nepal?

Line 114: How was height measured? All other measures have procedures stated.

Line 118: Manufacturer details missing

Line 126: Typo – ‘sport’ urine testing

Line 131: How were the urine sample containers provided to participants, i.e., were they delivered in person by the researchers the night before the interview? What is the relevance of the QR code in this context? Were all participants requested to submit a urine sample?

Line 132: Was a minimum level of urine required/specified for collection?

Line 136: Please clarify what you mean by ‘nearly located places’

Line 140-141: I’m not sure what you mean by this sentence. Please clarify how you managed pregnant women with regards to these measurements.

Line 141-143: Please clarify your rationale for excluding participants above and below these criterion values.

Line 173-174: What do you mean by the salt and non-salt participants?

Line 178-179: More detail is needed on the statistical tests in relation to specific variables of interest (e.g., how many factor ANOVA, what groups, etc?), and also clear rationale for using the selected methods (e.g., why both ANOVA and multivariable/multivariate regression?). Were follow-up tests employed where differences were noted? This needs more clarity here and in the results section.

Line 180: The terms multivariate and multivariable regression are used interchangeably in this paper. Please clarify the statistical methods employed. You may wish to refer to this article for reference: https://www.ncbi.nlm.nih.gov/pmc/articles/PMC3518362/

Line 185: This implies only verbal study information was provided to prospective participants. Was written information not supplied also? If so, were participants given ample time to consider their involvement / seek advice from others / ask any questions to the study team? Please clarify these points.

Results:

Line 201: Why was one PSU “dropped”?

Line 203: An incorrect percentage is reported for the sample attained at Step 3.

Line 204: You refer to consent for the urine test. Was consent for this aspect sought independently from ‘general’ consent for participation? If so, this needs clarifying earlier in the methods section.

Line 231: Clarify what you mean by ‘SE’ here. Why are you reporting standard error (assuming this is what you mean) and not standard deviation?

Line 247: Is this per day?

Line 249-250: If this result is not significant in table 3, then reasonably you cannot suggest an association between education level and salt intake based upon your data.

Line 251: As above, this result was non-significant in table 3. It is misleading to suggest that wealth is impacting on salt intake based upon this data. However, I note in table 4 that a significant result is reported between the highest and lowest wealth index. Therefore, you need to be clear what the information in this paragraph is referring to, e.g., table 3 or table 4 data.

Line 255: What does “(0.19)” refer to following BMI at this point?

Lines 255-257: Data in table 3 shows that there was no significant difference between hypertensive and normotensive participants, yet here you state that salt intake differed. Again, is this related to data in table 3 or table 4?

NB. It appears that data in tables 3 and 4 are conflicting at times, for certain variables, and this means that the reader may be confused by the presentation of your findings at this stage of the manuscript. Further justification and explanation is warranted as to why the specific data analyses have been used, and the merit of each method discussed.

Table 1:

a) A number of measures are reported in this table that are not mentioned in the methods section. Please refer to all assessed measures in the methods section.

b) Mean ± SD age should not be reported in column 1. Consider incorporating a ‘total’ column that would allow this data to be presented more appropriately.

c) Data for age for men and women are confusing with the units used. SI units for age need to be reported. The use of ‘n’ implies that you’re reporting 42.2 males, when in fact I believe you’re reporting that the mean age for males was 42.2 y. Please clarify.

d) Similarly, SI units need to be reported for age ranges.

e) SI units in column headings suggest that percentages will be reported in parentheses, but this is only done for the first factor, age. All other values are missing.

f) How has residence been categorised? What are the definitions / criteria used to determine whether someone resides in a metropolitan/sub-metropolitan, municipality, or rural area? This information needs to be included within the methods section.

g) Check table for formatting inconsistencies (e.g., bold)

h) How has wealth been categorised? What are the definitions / criteria used to determine the lowest, second, middle, fourth, highest category of wealth? This information needs to be included within the methods section.

i) Consider splitting the sociodemographic information from the anthropometric and metabolic variables within this table, either by using two separate tables or at least sub-dividing the data – particularly as these measures have been clearly delineated into Step 1, 2 and 3 of data collection.

j) Reporting of anthropometric measures is confusing, again due to the inconsistent way units are being reported. I suggest that you remove the reference to mean ± SD from column 1, where used, and replace this with the appropriate SI units for each variable, e.g., height (cm), weight (kg), etc. Then, where applicable, use an asterisk on the data in column 1 where values will be being reported as mean ± SD in columns 2 and 4, and write a simple sentence explaining this underneath the table. Use this method for other variables, where applicable. Units should then be removed from the column headings of columns 2 and 4.

k) Waist circumference is reported but not hip circumference, which was stated in the methods section. This needs to be included. I would also like to see the waist to hip ratio calculated for ease of comparison across studies.

l) It is unclear what values are being reported in columns 2 and 4 for the sub-categories of BMI. Please clarify.

m) What do you mean by “BP (high)”? These data need clarifying, and all abbreviations for blood pressure need defining at first use.

n) Please clarify what you mean by “people measured to be hypertensive on treatment”. Is data reported an absolute number? What do you mean by ‘treatment’, e.g., pharmacotherapy, lifestyle management, etc? If pharmacotherapy, were differing medication / regimes documented? Further details are needed on this in the methods section.

o) Please clarify your variable “Diabetes mellitus”. Is this Type 2, specifically, or all diagnoses of diabetes mellitus? Units need clarifying.

p) What do you mean by “cholesterols (high)”? What type of cholesterol was measured? Further detail needed in the methods section.

q) What do you mean by “Ever had a heart attack or chest pain from heart disease or stroke”? This question is confusing. How would people know that a symptom of chest pain was a result of heart disease, and what is the link here with stroke? Further clarity is needed in the methods section on this aspect.

Table 2:

a) Line 1, column 1 appears out of line.

b) Values reported for salt intake greater than 10% above the WHO recommendation differ in the table (31%) compared to what is stated in the paragraph above and abstract (48.9%). The latter value is reported in the table for potassium intake. Please clarify these data. Also, data reported in a table should not be reported in written text, to avoid duplication.

c) Standard error for total sodium/potassium ratio has been omitted.

Table 3:

a) SI units need to be reported.

Discussion:

Line 263: I’m not sure that I agree with this assertion. I do not feel that the primary aim of this study was to evaluate the capacity of spot urine samples for estimating mean population salt intake in Nepal. One of your aims was to estimate population salt intake using spot urine samples; you did not provide an “in depth evaluation” of this method in this study, however.

265-268: Make it clear here if you are referring to data in your study or elsewhere.

Line 280: Why might the spot urine sample be providing lower values than 24h urine measurements? You have made the comparison on methods and introduced a discrepancy in the findings, but do not provide any reasoned argument as to why this might be or whether one method might be more valid or reliable? Please comment.

Line 305: You state this finding is “not surprising”, as it has been evidenced before in a previous study, but why might salt increase be decreasing with advancing age?

Line 313: Was salt intake “low” or lower in those with higher education level? To my knowledge, they still had higher salt intake than the recommended level.

Line 318: Is there any suggestion as to why these provinces might be consuming more salt than others? It seems the underlying reasons for this were not assessed in this study, but perhaps there is evidence elsewhere that could provide insight, or maybe this might be a proposed area for future research?

Line 368: I do not disagree with this statement, but it is pitched as if your study has demonstrated this fact, when actually no cost-effectiveness analyses have been incorporated. Consider revising this.

6. PLOS authors have the option to publish the peer review history of their article (what does this mean?). If published, this will include your full peer review and any attached files.

Reviewer #1: No

Reviewer #2: No

---

## [Author Response · Author response to Decision Letter 0]

19 Jun 2021

06 June 2021 

Shyam Sundar Budhathoki, MBBS, MD, MPH

Academic Editor

PLOS ONE

PONE-D-21-06676

Estimation of mean Population salt intakes using Spot Urine samples and association with Body Mass Index, Hypertension, Raised Blood Sugar and Blood Lipids: Findings from the Non-communicable Disease Risk Factors (STEPS) Survey 2019 in Nepal

Dear Dr. Budhathoki

Thank you very much for your email of 23 April 2021 and comments on our manuscript. We have carefully revised the manuscript in response to the extensive and insightful comments we received from you and reviewer.

In particular, the reviewer provided constructive comments with some corrections. In the revised manuscript version, all the suggested corrections are made. We have also revised the content of introduction, methodology, result and discussion section according to reviewers’ advice and have paid special attention to correcting all typological errors. Appended below is the list of all yours and reviewer comments along with our responses to each point.

We hope that this revised version will be suitable for publication in PLoS ONE. 

Yours sincerely,

Meghnath Dhimal 

Journal Requirements:

 ** Thank you so much for your suggestions and we have carefully formatted our manuscript as per Journal requirements. 

 ** Thank you so much for your suggestions. The data of this study is available from the WHO NCD Microdata Repository (https://extranet.who.int/ncdsmicrodata/index.php/catalog/771/data_dictionary )

3. Thank you for submitting the above manuscript to PLOS ONE. During our internal evaluation of the manuscript, we found significant text overlap between your submission and the following previously published works, some of which you are an author.

- https://www.bmj.com/content/bmj/368/bmj.m315.full.pdf

- https://onlinelibrary.wiley.com/doi/full/10.1111/jch.12778

Please revise the manuscript to rephrase the duplicated text, cite your sources, and provide details as to how the current manuscript advances on previous work. Please note that further consideration is dependent on the submission of a manuscript that addresses these concerns about the overlap in text with published work.

 ** Thank you so much for your suggestions and we have rephrased/re-written all indicated texts. 

Review Comments to the Author 

Reviewer #1: This study aims to determine the population mean of salt intakes and its correlates using 2019 STEPS survey data in Nepal. The research question is adequately motivated. However, there are some concerns that authors need to address to improve the quality of the manuscript.

Comments

• The title and objectives of the study are not consistent.

** Thank you for your suggestion. Necessary changes has been made in revised manuscript.

• Title needs some improvement. For example, while hypertension and raised blood sugar indicate a direction, the authors did not mention about BMI and lipid level similarly. Please revise the title. Also, use hypercholesterolemia instead of high cholesterol level.

** Thank you for your suggestion and instructions; the title has been revised to “Estimation of mean population salt intakes using spot urine samples and associations with body mass index, hypertension, raised blood sugar and hypercholesterolemia: Findings from Non-communicable Disease Risk Factors (STEPS) Survey 2019 in Nepal”.

• It is highly preferable to use consistent terms throughout the whole manuscript. For instance, raised blood sugar is used in the title but there is no indication of such variables in the result, except diabetes.

** Thank you for your suggestion. We have included the term “raised blood sugar” into the revised manuscript where appropriate.

• The authors mentioned that the equation from INTERSALT was developed using a large heterogenous population sample. Do the authors mean by the term “heterogenous” that the equation can be applied to the Nepalese population?

** Thank you for your feedback.

We have mentioned in the discussion section in Line: 757-758, that it is one of the limitations of our study. Similarly, 24-hours urine is gold standard method for estimating mean population salt intake; however, estimation of salt intake based upon spot urine samples elsewhere had shown excellent concordance with 24-hr urine collection measurements and had outstanding sensitivity (97%) and specificity (100%) at classifying mean population salt intake as above or below the World Health Organization maximum guideline value of 5 g/day, which is mentioned in the discussion section, Line: 577-580. Likewise: We have mentioned in the Introduction section in line 143-145 that for a country to address an estimate of mean national salt intake and, in December 2013, the WHO incorporated measurement of mean population salt intake as an element of the WHO Stepwise approach to Surveillance (STEPS) protocol. 

• The authors also explicitly mentioned the equation INTERSALT for southern Europe. If this is the case, there would be at least some anthropometric differences between these and Nepalese population? For example, considering BMI and age adjustment to the equation would seem to be more appropriate instead of using the original equation.

** Thank you for your feedback. We have not validated these equations in our local context. 

However, after discussion with relevant local experts and WHO expert teams, we decided to use the equation for southern Europe (INTERSALT) for estimation of daily salt intake. We have mentioned in the methodology section in line: 332-334 that, so far there is no consensus on the equation to be used in a given population/context. The estimation in this survey maintained the use of the same equation as in previous survey rounds to facilitate comparison of results and assessment of trends

• In Table 2, the prevalence of salt intake > 5 g was higher in women than in man, whereas that of salt intake > 10 g was reversed, higher in man than in women. Please discuss this.

**Thank you for your suggestion. We have added the following text in our discussion section (Line: 349-351).

“This study shows that the prevalence of salt intake > 5 g was higher in females than in males, whereas that of salt intake > 10 g was reversed, higher in males than in females. These findings might be linked to different lifestyle and behavioral choices between the male and female population including women being less likely to exhibit extremes in selection of both type and amounts of food, .

• As the authors rightly discussed that people with history of hypertension may be aware of their hypertensive status and likely to reduce salt intake. This can also be linked to behavioral and lifestyle choices. That’s why salt intake is higher in men as women are more likely to take care of themselves. Likewise, BMI can also be attributed by lifestyle choices and hence, by salt intake. Therefore, there is no clear indication of exposure and outcome pathway in this model.

**Thank you for your suggestion. We have added and deleted the necessary text in our discussion section as documented below: .

Lines 380-383: “One of the possible reasons for increased High sodium intake as BMI increased has been suggested as an indirect cause of obesity in this study might be attributed to unhealthy lifestyles, including poor diet, physical inactivity, and sedentary behaviors of the study population”.

Line 393-394: “However, our finding reported salt intake was less among the hypertensive population compared to non-hypertensive. This may be due to reverse causation but as our study is cross-sectional, it is not possible to test this hypothesis”.

• What are the clinical implications of the study beyond the assessment of population salt intake and validation of equation in Nepalese population?

**Thank you for your suggestion. We have added following text in our discussion section (Line: 897-903). 

“This study has important clinical and public health implications. In Nepal, almost 24.5% of the Nepalese population had raised blood pressure, and more than 22% of NCDs deaths are related to CVDs [8]. There is strong and consistent evidence that excess salt intake remains one of the key risk factors for high BP and thereby increased risks of CVDs, and stroke [3]. Effective interventions to reduce salt intake at the population levels present an opportunity to combat the increasing burden of NCDs and increase life expectancy.”

• Conclusion should be based on the study findings. For instance, the study’s results did not present the lifestyle modification and its importance on health indicators. It is also not a finding of the study that “population-based salt reduction strategies are cost-effective and cost-saving in most of the settings for prevention of non-communicable diseases.”. Please revise the conclusion.

**Thank you for your suggestion. We have revised the conclusion accordingly (Line: 914-922). 

“This study demonstrated the association of high salt intake with sex, age range, education, province, excess weight, and cholesterol level of participants. The difference in salt intake showed a positive relationship with excess weight and a negative relationship with biochemical risk factors such as hypertension, raised blood sugar, and cholesterol levels. Our study clearly highlighted the need for future studies using longitudinal data or randomized clinical trials to assess the role of dietary salt in the development/prevention of blood pressure, raised bold sugar and cholesterol levels in the Nepalese population. Furthermore, these findings build a strong case of actions needed to reduce salt consumption in Nepal to achieve the global target of a 30% reduction in population salt intake by 2025.”

• References should be relevant and up-to-date. Please try to reduce the number of references.

**Thank you for your suggestion. Necessary changes has been made in References section in the revised manuscript.

• in Table 4, there is no need to report R squared or adjusted R squared without presenting all models. Remove the last two columns.

• If p-value is 0.000, indicate using “<0.001” in Table 3 and 4.

**Thank you for your suggestion. We have removed R-squared values from the table and changed p-values of 0.000 to “<0.001”.

Reviewer #2: General Comments:

This study used data collected from spot urine tests to estimate population-level salt intake in Nepal, and to assess the relationship of salt intake with various anthropometric and metabolic factors. Data was derived from the well-known STEPS Survey and incorporated a large, nationally-representative study population. The study has potentially very relevant and important findings for future intervention for NCDs in Nepal. However, I have a number of concerns and/or queries with some of the information being presented in its current form, which I would like the authors to address.

Firstly, in the introduction, there needs to be more emphasis as to why salt intake corresponds to various anthropometric and metabolic variables, so as to build a strong rationale for the investigation into these relationships in this study. Particular attention should be paid to the variables that are subsequently stated in the study aim.

**Thank you for your constructive feedback. We have emphasized associations between excessive salt intake and CVD risk factors (as measured in our study) as follows: 

Lines 63-66: ” In addition to hypertension, there is also evidence of associations between excessive salt consumption to be indirectly related to obesity and other CVD risk factors such as diabetes and hypercholesteremia.”

Greater clarity is needed over the statistical methods employed in this study, as at times important data appear to have conflicting findings depending on the analyses used. See specific comments in results section.

**Thank you for your feedback. Following changes have been made in revised manuscript. 

“Multivariable linear regression was used to estimate the differences in salt intake in relation to different explanatory variables such as age category, sex, province, education, income, BMI, and blood pressure, raised blood sugar and cholesterol level.”

Tables need revising for better clarity on the reporting of data, particularly concerning SI units.

**Thank you for your feedback. Necessary changes have been made Results section in revised manuscript.

“ SI units are inserted for given variables for better clarity, Tables were revised as per suggestion (Breakdown of Tables 1 was made for Steps 1,2 and 3, waist and hip ratio was added in Table 1, mean and SD was presented as per instruction for all the tables) for better clarity. 

The discussion covered the major points raised in the study. However, clear assessment of this section is hampered at this time by outstanding questions over the presentation (and conflicts) of results using ANOVA and multivariable/multivariate regression. Important limitations were highlighted in the discussion, which was good to see, as well as the study strengths. I feel you could make the strengths of this study seem more impactful though, by concentrating less on it being the “first” study to show these findings and more on the relevance, i.e., that it is a Nationwide population study, that incorporated individuals of all regions, caste, sex and age, and thus the study findings are highly generalisable for Nepal. This is an important step in determining future interventions to reduce salt intake among Nepalese peoples.

**Thank you for your constructive suggestion. The following text has been added on the manuscript. 

“To our best knowledge, this is the first nationwide population study to estimate mean population-level salt intake using spot urine samples in the Nepalese population. This study has several strengths, at first, it is a nationwide population study, that incorporated individuals of all regions, caste, sex, and age, and thus the study findings are highly generalizable for Nepal. Second, to our best knowledge, this is the first study in the Nepalese population, which demonstrated the association of salt intake with different characteristic. This is an important step in determining future interventions to reduce salt intake among Nepalese population,”

Although I commend the authors on a generally well-written manuscript, there were several grammatical errors noted throughout and, particularly as PLOS One do not copy edit, the authors should ensure that the written language is reviewed and modified accordingly.

Thank you for your feedback and suggestions. The revised version of our manuscript has been proof-read and edited by an epidemiologist who is a native English speaker.

Specific Comments:

Introduction:

Line 87: You state equations have been investigated in several studies but provide no indication of the validity or reliability of these methods. Please expand further by discussing the strengths, limitations and validity and reliability of using estimation methods.

** Thank you for your feedback and suggestions. We have mentioned in discussion sections (Line: 76-580) about the validity and reliability of these methods as follows. 

“The standard approach to measuring the mean salt intake of a population has been the collection of 24-h urine samples on a subset of individuals. However, this method is troublesome, time-consuming, costly to participants due to the complex nature of urine sample collection and may miss the sodium estimation excreted through non-urinary routes and some dietary sodium derives from sources other than salt. Equations that use spot urine samples to estimate population salt intake have been explored as a possible alternative in a number of studies. WHO included spot urine in STEPs survey protocol in December 2013, as a measure to estimate mean population salt intake”.

“Estimation of salt intake based upon spot urine samples elsewhere had shown excellent concordance with 24-hr urine collection measurements [27] and had outstanding sensitivity (97%) and specificity (100%) at classifying mean population salt intake as above or below the World Health Organization maximum guideline value of 5 g/day.”

Line 88-89: The grammar isn’t quite right here – I’m not sure what you’re meaning.

**Thank you for your suggestion. Necessary changes have been made in revised manuscript (Line: 173-180). 

Original

“For a country to address an estimate of mean national salt intake and, in December 2013, the WHO incorporated measurement of mean population salt intake as an element of the WHO Stepwise approach to Surveillance (STEPS) protocol.”

Changed to:

“WHO in 2013 include spot urine in STEPs survey protocol, as a measure to estimate mean population salt intake”

Line 91-: You need to be clearer on your aim/purpose. Use future tense (what you aim to do) rather than past tense (what you have already done).

** Thank you for your suggestion. Following changes have been made in revised manuscript (Line: 180-184). 

“Hence; we collected spot urine samples from a nationally representative population of Nepal for the first time to estimate the mean population salt intake. In this study we also evaluated the associations of salt intake with cardiovascular disease risk factors including excess weight (BMI), hypertension, raised blood sugar and cholesterol levels as well as their relationships with socio-demographic characteristics”. 

Methods:

Line 102: Was the sample of 925 per province selected in advance, i.e., was this a target? If so, why was this particular number selected? Was there a margin, or mechanism in place in case you weren’t able to recruit exactly this number (i.e., it seems as though only this precise number of households were contacted, but what if they refused to participate in the survey)?

** Thank you for your suggestion. Sample size was calculated using WHO STEPS sample size calculator. For non-participants it was mentioned as non-response. The changes suggested have been made in revised manuscript (lines:194).

Line 104: Could you provide further clarity on what constitutes a primary sample unit. How were these identified?

** Thank you for your suggestion. For the sampling process, the ward was considered as the primary sampling unit (PSU), which I have mentioned, in revised manuscript (Line: 196).

Line 107: Please clarify what you mean when you say ‘using the android tablet’. Is this specifically in relation to randomly sampling one adult in each household, or something else? In what way was the android tablet used?

** Thank you for your feedback. Android tablets were used for all data collection process, as well listing of particular households, than it was used to randomly assigned eligible members for study purpose. For ease understanding I have revised the statements in the manuscript as follows:

“The survey listed all the households of 259 primary sample units (PSUs) using probability proportionate to size (PPS), 37 PSUs in each of 7 Provinces, 25 households per PSU were sampled using systematic random sampling. From each of the selected households, one eligible participant (15–69 years) from selected household was chosen randomly and approached for the study”

Line 110 onwards: If relevant, please clarify that the field research assistants led/facilitated all or which parts of the data collection (i.e., steps 1-3?). Please also clarify the stated background experience, i.e., presumably, researchers were required to have one of the stated backgrounds rather than expecting to have experience in all of these fields?

** Thank you for your suggestion. Necessary changes have been made in revised manuscript (Line: 267). 

“A total of 60 field research assistants in health sciences backgrounds were mobilized all over Nepal”.

Line 113: Further detail is needed on the interview process. How was this conducted? Did the researchers ask questions and complete the answers for the participants, or were they just ‘on-hand’ to answer any questions if they arose whilst the participants completed the questionnaires? Were consistent methods applied throughout Nepal?

** Thank you for your suggestion. Researchers asked questions and completed the answers for the participants. This has been added at 268-270.

“

Line 114: How was height measured? All other measures have procedures stated.

** Thank you for your suggestion. Necessary changes have been made in revised manuscript.

“Height and weight were measured with a portable digital weighing scale (Seca, Germany).” 

Line 118: Manufacturer details missing

** Thank you for your suggestion. We have mentioned the manufactured details in line (274-280)

Line 126: Typo – ‘sport’ urine testing

** Thank you – we have corrected this misspelling. 

Line 131: How were the urine sample containers provided to participants, i.e., were they delivered in person by the researchers the night before the interview? What is the relevance of the QR code in this context? Were all participants requested to submit a urine sample?

** Thank you for your feedback. Following changes have been made in revised manuscript (Line: 283-286). 

“For a urine collection, urine container was provided to each participant to collect spot urine after completing the Steps 1 and 2 data collection. The instruction for spot urine collection was given and asked them to bring the urine sample with them to the appointment for blood testing the next morning.”

Line 132: Was a minimum level of urine required/specified for collection?

** Thank you for your feedback. he minimum level of urine required was not mentioned but the participants were provided 10 ml urine containers and asked them to return them as full as possible. This had been added to the revised manuscript at lines 304. 

Line 136: Please clarify what you mean by ‘nearly located places’

** Thank you for your feedback. We have revised the sentence in manuscript (Line:262) as follows:

“Laboratory setup was done in every province headquarters for analysis of urine collected for that province.”.

Line 140-141: I’m not sure what you mean by this sentence. Please clarify how you managed pregnant women with regards to these measurements.

** Thank you for your feedback. While collecting socio-demographic information’s in Steps-1, pregnant females were excluded from providing a urine sample. This has been added at lines 314-317 in the revised manuscript.

Line 141-143: Please clarify your rationale for excluding participants above and below these criterion values.

** Thank you for your feedback. Criteria were set in the STEPs survey protocol to reduce the bias when generalizing the mean salt intake for the general population. 

Line 173-174: What do you mean by the salt and non-salt participants?

** Thank you for your suggestion. The change was made as follows for ease understanding. (Line: 355-356) 

“Descriptive statistics for demographics and collected data were recorded for all participants who consented to Steps 1 of the survey.”

Line 178-179: More detail is needed on the statistical tests in relation to specific variables of interest (e.g., how many factor ANOVA, what groups, etc?), and also clear rationale for using the selected methods (e.g., why both ANOVA and multivariable/multivariate regression?). Were follow-up tests employed where differences were noted? This needs more clarity here and in the results section.

** Thank you for your suggestion. The statement was revised as follows for better clarity (Line: 360-362) 

“Student's ‘t’ test, one-way ANOVA, and Chi-square statistics were used to determine the association of mean salt consumption levels with explanatory factors. The relationship between salt intake and participant characteristics was explored using multivariable linear regression.” 

Line 180: The terms multivariate and multivariable regression are used interchangeably in this paper. Please clarify the statistical methods employed. You may wish to refer to this article for reference: https://www.ncbi.nlm.nih.gov/pmc/articles/PMC3518362/

** Thank you for your suggestion. Necessary changes have been made in revised manuscript as follows:

“Multivariable linear regression was used to estimate the differences in salt intake in relation to different explanatory variables such as age category, sex, province, education, income, BMI, and blood pressure, raised blood sugar and cholesterol level.”

Line 185: This implies only verbal study information was provided to prospective participants. Was written information not supplied also? If so, were participants given ample time to consider their involvement / seek advice from others / ask any questions to the study team? Please clarify these points.

** Thank you for your suggestion. Necessary changes have been made in revised manuscript as follows:. (Line: 375-378) 

“Written informed consent was obtained from each of the research participants jointly for Steps 1&2 and for steps 3. Participants were informed regarding their right to withdraw from the study at any time without penalty and issues concerning confidentiality and consent will be upheld in accordance with ethical research standards.” 

Results:

Line 201: Why was one PSU “dropped”?

** Thank you for your suggestion. We have mentioned it in revised manuscript as follows: (Line: 405).

“Amongst the initially planned 6475 sample size, 1 PSU with 25 participants was dropped due to heavy snow, leaving 6450 as our total sample size.”

Line 203: An incorrect percentage is reported for the sample attained at Step 3.

** Thank you for your suggestion. We have corrected this percent in the revised manuscript (Line: 408). 

Line 204: You refer to consent for the urine test. Was consent for this aspect sought independently from ‘general’ consent for participation? If so, this needs clarifying earlier in the methods section.

** Thank you for your suggestion. We have not sought the consent differently for urine collection; We have mentioned the consent procedure in line 375-378 with better clarity in revised manuscript. 

Line 231: Clarify what you mean by ‘SE’ here. Why are you reporting standard error (assuming this is what you mean) and not standard deviation?

** Thank you for your suggestion. Necessary changes have been made in Results section in revised manuscript. As per feedback of reviewer, uniformity has been made in the revised manuscript. We only reported standard deviation (SD). 

Line 247: Is this per day?

Yes, It was per day. This has been clarified in the revised manuscript (line: 584).

Line 249-250: If this result is not significant in table 3, then reasonably you cannot suggest an association between education level and salt intake based upon your data.

** Thank you for your suggestion. Though it was not significant in table 3, we were trying to find out the differences of salt intake among participants in relation to education level, We have revised the findings in revised manuscript as (Line: 586),

 “Participants who had more than secondary education ate less salt compared to participants who had primary and secondary education in revised manuscript.”

Line 251: As above, this result was non-significant in table 3. It is misleading to suggest that wealth is impacting on salt intake based upon this data. However, I note in table 4 that a significant result is reported between the highest and lowest wealth index. Therefore, you need to be clear what the information in this paragraph is referring to, e.g., table 3 or table 4 data.

Line 255: What does “(0.19)” refer to following BMI at this point?

** Thank you for your suggestion. Here, we tried to access the differences in salt consumption level in relation to wealth index. We have revised the text in revised manuscript (Line: 588) as

 “Similarly, salt intake decreases as the wealth index of the participant's increases”.

Lines 255-257: Data in table 3 shows that there was no significant difference between hypertensive and normotensive participants, yet here you state that salt intake differed. Again, is this related to data in table 3 or table 4? It referred to Table 4. After your suggestion it has been revised for better clarity as 

“Participants who were hypertensive, and had raised blood cholesterol ate less salt than people who had normal blood pressure and cholesterol level”

NB. It appears that data in tables 3 and 4 are conflicting at times, for certain variables, and this means that the reader may be confused by the presentation of your findings at this stage of the manuscript. Further justification and explanation is warranted as to why the specific data analyses have been used, and the merit of each method discussed.

** Thank you for your suggestion. After your constructive feedback, we have revised the manuscript and mentioned the aims of performing different statistical analysis in methods section as follows:.

“Student's ‘t’ test, one-way ANOVA, and Chi-square statistics were used to determine the association of mean salt consumption levels with explanatory factors. The relationship between salt intake and participant characteristics was explored using multivariable linear regression. Multivariable linear regression was used to estimate the differences in salt intake in relation to different explanatory variables such as age category, sex, province, education, income, BMI, and blood pressure, raised blood sugar and cholesterol level.”

Table 1:

a) A number of measures are reported in this table that are not mentioned in the methods section. Please refer to all assessed measures in the methods section.

b) Mean ± SD age should not be reported in column 1. Consider incorporating a ‘total’ column that would allow this data to be presented more appropriately.

c) Data for age for men and women are confusing with the units used. SI units for age need to be reported. The use of ‘n’ implies that you’re reporting 42.2 males, when in fact I believe you’re reporting that the mean age for males was 42.2 y. Please clarify.

d) Similarly, SI units need to be reported for age ranges.

e) SI units in column headings suggest that percentages will be reported in parentheses, but this is only done for the first factor, age. All other values are missing.

f) How has residence been categorised? What are the definitions / criteria used to determine whether someone resides in a metropolitan/sub-metropolitan, municipality, or rural area? This information needs to be included within the methods section.

g) Check table for formatting inconsistencies (e.g., bold)

h) How has wealth been categorised? What are the definitions / criteria used to determine the lowest, second, middle, fourth, highest category of wealth? This information needs to be included within the methods section.

i) Consider splitting the sociodemographic information from the anthropometric and metabolic variables within this table, either by using two separate tables or at least sub-dividing the data – particularly as these measures have been clearly delineated into Step 1, 2 and 3 of data collection.

j) Reporting of anthropometric measures is confusing, again due to the inconsistent way units are being reported. I suggest that you remove the reference to mean ± SD from column 1, where used, and replace this with the appropriate SI units for each variable, e.g., height (cm), weight (kg), etc. Then, where applicable, use an asterisk on the data in column 1 where values will be being reported as mean ± SD in columns 2 and 4, and write a simple sentence explaining this underneath the table. Use this method for other variables, where applicable. Units should then be removed from the column headings of columns 2 and 4.

k) Waist circumference is reported but not hip circumference, which was stated in the methods section. This needs to be included. I would also like to see the waist to hip ratio calculated for ease of comparison across studies.

l) It is unclear what values are being reported in columns 2 and 4 for the sub-categories of BMI. Please clarify.

m) What do you mean by “BP (high)”? These data need clarifying, and all abbreviations for blood pressure need defining at first use.

n) Please clarify what you mean by “people measured to be hypertensive on treatment”. Is data reported an absolute number? What do you mean by ‘treatment’, e.g., pharmacotherapy, lifestyle management, etc? If pharmacotherapy, were differing medication / regimes documented? Further details are needed on this in the methods section.

o) Please clarify your variable “Diabetes mellitus”. Is this Type 2, specifically, or all diagnoses of diabetes mellitus? Units need clarifying.

p) What do you mean by “cholesterols (high)”? What type of cholesterol was measured? Further detail needed in the methods section.

q) What do you mean by “Ever had a heart attack or chest pain from heart disease or stroke”? This question is confusing. How would people know that a symptom of chest pain was a result of heart disease, and what is the link here with stroke? Further clarity is needed in the methods section on this aspect.

Table 1 has been revised as suggestion in revised manuscript. We have removed the variable ever had a heart attack in the revised table, since we were not taking consideration of these variables because response for these variables was very low to used for further analysis. 

Table 2:

a) Line 1, column 1 appears out of line.

b) Values reported for salt intake greater than 10% above the WHO recommendation differ in the table (31%) compared to what is stated in the paragraph above and abstract (48.9%). The latter value is reported in the table for potassium intake. Please clarify these data. Also, data reported in a table should not be reported in written text, to avoid duplication.

c) Standard error for total sodium/potassium ratio has been omitted.

Thank you, we have revised and corrected the percentage as well as we have added the SD for sodium/potassium ratio. 

Table 3:

a) SI units need to be reported.

Thank you for your suggestion. SI units have been added in Table 3. 

Discussion:

Line 263: I’m not sure that I agree with this assertion. I do not feel that the primary aim of this study was to evaluate the capacity of spot urine samples for estimating mean population salt intake in Nepal. One of your aims was to estimate population salt intake using spot urine samples; you did not provide an “in depth evaluation” of this method in this study, however.

** Thank you for your constructive feedback. Necessary changes have been made in revised manuscript as follows: (Line: 669-670). 

“To our best knowledge, this is the first nationwide population study to estimate mean population-level salt intake using spot urine samples in the Nepalese population.” 

265-268: Make it clear here if you are referring to data in your study or elsewhere.

** Thank you for your suggestion. We have rephrased the sentence in the revised manuscript as follows: (Line: 671). 

“Estimation of salt intake based upon spot urine samples elsewhere had shown excellent concordance with 24-hr urine collection measurements and had outstanding sensitivity (97%) and specificity (100%) at classifying mean population salt intake as above or below the World Health Organization maximum guideline value of 5 g/day.” 

Line 280: Why might the spot urine sample be providing lower values than 24h urine measurements? You have made the comparison on methods and introduced a discrepancy in the findings, but do not provide any reasoned argument as to why this might be or whether one method might be more valid or reliable? Please comment.

** Thank you for your suggestion. I have mentioned in the discussion section of manuscript that 24-hours urine collection is gold standards for estimating mean salt intake. This has also been included as a limitations for our study (Line: 856-857).

Line 305: You state this finding is “not surprising”, as it has been evidenced before in a previous study, but why might salt increase be decreasing with advancing age?

** Thank you for your suggestion. Necessary changes have been made in revised manuscript as follows: (Line: 746-749). 

“There is no previous study available in Nepal to support these results but from the above findings, it can be inferred that middle and older age groups population are more likely to experience one or more NCDs in their life time and are likely to be aware of salt intake and hence practice and maintain healthier dietary practices.”

Line 313: Was salt intake “low” or lower in those with higher education level? To my knowledge, they still had higher salt intake than the recommended level.

** Thank you for your suggestion. Though, the mean salt intake was higher than WHO recommendation, we aim to sought the difference of salt intake among the participants with different education level. Hence, we just tried to emphasize the difference of mean salt intake with education level. 

Line 318: Is there any suggestion as to why these provinces might be consuming more salt than others? It seems the underlying reasons for this were not assessed in this study, but perhaps there is evidence elsewhere that could provide insight, or maybe this might be a proposed area for future research?

Thank you for your constructive feedback and suggestions. I have made the necessary changes in revised documents as follows: (Line: 762-765).

“Our study showed people who reside in Karnali and Sudhurpashchim province ate more salt than other provinces. There is no evidence in Nepal to support these results; one of the possible reasons could be a different dietary pattern and dietary habit of people of these provinces. This might be a proposed area of future research to validate these findings.”

Line 368: I do not disagree with this statement, but it is pitched as if your study has demonstrated this fact, when actually no cost-effectiveness analyses have been incorporated. Consider revising this.

** Thank you for your constructive feedback and suggestion. We have removed the statement and revised the conclusion accordingly. (Line: 920-922).

“Furthermore, these findings build a strong case of actions needed to reduce salt consumption in Nepal to achieve the global target of a 30% reduction in population salt intake by 2025.”

---

## [Decision Letter · Decision Letter 1]

3 Aug 2021

PONE-D-21-06676R1

Estimation of mean population salt intakes using spot urine samples and associations with body mass index, hypertension, raised blood sugar and hypercholesterolemia: Findings from Non-communicable Disease Risk Factors (STEPS) Survey 2019 in Nepal

PLOS ONE

Dear Dr. Dhimal,

Thank you for submitting your manuscript to PLOS ONE. After careful consideration, we feel that it has merit but does not fully meet PLOS ONE’s publication criteria as it currently stands. Therefore, we invite you to submit a revised version of the manuscript that addresses the points raised during the review process.

Detailed comments from the reviewer can be found in the attached PDF as annotations. In addition to the comments, please revise the manuscript for academic English and grammar, if possible with the help of a native user of English.

We look forward to receiving your revised manuscript.

Kind regards,

Shyam Sundar Budhathoki

Academic Editor

PLOS ONE

Reviewers' comments:

Reviewer's Responses to Questions

**Comments to the Author**

1. If the authors have adequately addressed your comments raised in a previous round of review and you feel that this manuscript is now acceptable for publication, you may indicate that here to bypass the “Comments to the Author” section, enter your conflict of interest statement in the “Confidential to Editor” section, and submit your "Accept" recommendation.

Reviewer #3: (No Response)

2. Is the manuscript technically sound, and do the data support the conclusions?

Reviewer #3: Partly

3. Has the statistical analysis been performed appropriately and rigorously? 

Reviewer #3: I Don't Know

4. Have the authors made all data underlying the findings in their manuscript fully available?

Reviewer #3: Yes

5. Is the manuscript presented in an intelligible fashion and written in standard English?

Reviewer #3: No

6. Review Comments to the Author

Reviewer #3: The key finding of this research is mean salt intake, process of data collection, discuss spot vs 24 hours more effectively, practicality of such test in developing country context and how this finding can inform public health measures in Nepal. I also think just discussing significant association with gender, obesity and others is enough; no need to stress about lack of association of salt intake with hypertension or diabetes (or identifying the need for RCT).

I see a lot of scope for improvement before this paper can be considered for publication. I have provided comments within the pdf paper as comments.

7. PLOS authors have the option to publish the peer review history of their article (what does this mean?). If published, this will include your full peer review and any attached files.

Reviewer #3: No

---

## [Author Response · Author response to Decision Letter 1]

4 Sep 2021

04 September 2021 

Shyam Sundar Budhathoki, MBBS, MD, MPH

Academic Editor

PLOS ONE

PONE-D-21-06676R1

Estimation of mean Population salt intakes using Spot Urine samples and association with Body Mass Index, Hypertension, Raised Blood Sugar and Blood Lipids: Findings from the Non-communicable Disease Risk Factors (STEPS) Survey 2019 in Nepal

Dear Dr. Budhathoki

Thank you very much for your email of 03 August 2021 and comments on our manuscript. We have carefully revised the manuscript in response to the extensive and insightful comments we received from the reviewer.

We noticed that the reviewer had commented on our originally submitted version and we have already addressed many comments on the revised manuscript (R1). In particular, the reviewer provided constructive comments with some corrections. In the revised manuscript version, all the suggested corrections are made. We have also revised the content of introduction, methodology, result and discussion section according to reviewers’ advice and have paid special attention to correcting all typological errors. Our revised manuscript is also edited by native speaker from UK. Appended below is the list of all yours and reviewer comments along with our responses to each point.

We hope that this revised version will be suitable for publication in PLoS ONE. 

Yours sincerely,

Meghnath Dhimal 

Review Comments to the Author 

1. May be running title reflect the main aim of the paper i.e. estimating mean population salt intake and associated factors...

** Thank you for your suggestion and instructions; the running title has been revised to “Estimating mean population salt intake and associated factors”. (Lines: 7)

2. Accurate measurement or population data or both??

**Thank you for your suggestion. We have revised the text accordingly. (Lines: 35).

3. What does for the first time mean? Is it "in STEPS survey or NHRC research or first time in Nepal??" May be remove it from abstract. Explain in main text.

**Thank you for your suggestion. Mean salt estimation was carried out in Nepal for the first time. We have remove the first time in abstract and mentioned it in main text.

4. I think the main purpose of this study was to estimate population level data to support public health strategies to reduce salt intake; not to establish a causal association of salt intake, right?

**Thank you for your constructive feedback and suggestion. We have removed the statement and revised the statement as follows. (Line: 59-61). 

“Furthermore, these findings build a strong case of actions needed to reduce salt consumption in Nepal to achieve the global target of a 30% reduction in population salt intake by 2025.”

5. Please simplify the Sentence as it is unclear may be in 2 sentences.

**Thank you for your suggestion. We have revised the text accordingly. (Lines: 74).

6. How does the disease put economic burden? Please add one or two sentence.

**Thank you for your suggestion. We have added the following text in the manuscript. (Lines: 88-90).

“NCDs incurred economic burden by increasing health-care costs, lowering workplace productivity, increasing sick days, and inflicting permanent disability”. 

7. I think this name of not complete action plan for what? Control of NCDs.

**Thank you for your constructive feedback. The sentence is revised in main text (Lines: 94-95).

8. I think the authors should focus on utility and practicality of spot collection/test…Challenges of 24h urine collection/test such as “may miss the sodium estimation excreted through non-urinary routes and also applies to spot collection/test; maybe remove irrelevant sentences. 

**Thank you for your feedback and suggestions. We had addressed those comments on revised manuscript (Line: 99-112). 

“The standard approach to measuring the mean salt intake of a population has been the collection of 24-h urine samples on a subset of individuals. However, this method is troublesome, time-consuming, costly to participants due to the complex nature of urine sample collection and may miss the sodium estimation excreted through non-urinary routes and some dietary sodium derives from sources other than salt. Equations that use spot urine samples to estimate population salt intake have been explored as a possible alternative in a number of studies. WHO included spot urine in STEPs survey protocol in December 2013, as a measure to estimate mean population salt intake”.

9. Why are these references not written as 21-28??

**Thank you for your suggestion. We have corrected the references. (Line: 111)

10. Methods 

** Thank you for your valuable and constructive suggestion and instructions in methodological section; we had addressed those comments on revised manuscript submitted after first revision, however; we received the comments on manuscript that was submitted at beginning. However; any missing revisions that were raised has been revised accordingly in methodological section. 

11. Was there any active attempt to identify how many were under hypertension medication or have received counseling from health workers and how many were taking their BP measurements for the first time? 

**Thank you for your query. We have collected all the information as mentioned above which can be found in Survey report. Link: https://www.who.int/ncds/surveillance/steps/2012-13_Nepal_STEPS_Report.pdf. 

12. I think just discussing significant association is enough; no need to stress about lack of association of salt intake with metabolic factors. The key aspect of this research is mean salt intake and how this could inform public health measures in Nepal. 

** Thank you for your suggestions and helpful criticism. We have handled such issues on amended manuscripts submitted after first revision, as previously stated; however, we had received comments on manuscripts submitted during the first round. Any missing revisions that were raised, however, have been addressed in both the discussion and conclusion sections. In conclusion section, we have revised the statement as follows. (Line: 453-459). 

“According to our findings, Nepal has a daily salt intake that is around double that of the WHO's recommended limit. A total of 70.8% of the population had a salt intake >5 g/d. The findings of this study will aid policymakers and other important stakeholders in Nepal at the central, federal, and local levels in tracking trends and progress toward salt intake targets, as well as guiding future policies and initiatives aimed at reducing salt intake among the population. Furthermore, these findings build a strong case of actions needed to reduce salt consumption in Nepal to achieve the global target of a 30% reduction in population salt intake by 2025”.

---

## [Decision Letter · Decision Letter 2]

25 Mar 2022

Estimation of mean population salt intakes using spot urine samples and associations with body mass index, hypertension, raised blood sugar and hypercholesterolemia: Findings from STEPS Survey 2019, Nepal

PONE-D-21-06676R2

Dear Dr. Dhimal,

We’re pleased to inform you that your manuscript has been judged scientifically suitable for publication and will be formally accepted for publication once it meets all outstanding technical requirements.

Kind regards,

Shyam Sundar Budhathoki

Academic Editor

PLOS ONE

Additional Editor Comments (optional):

Reviewers' comments:

Reviewer's Responses to Questions

**Comments to the Author**

1. If the authors have adequately addressed your comments raised in a previous round of review and you feel that this manuscript is now acceptable for publication, you may indicate that here to bypass the “Comments to the Author” section, enter your conflict of interest statement in the “Confidential to Editor” section, and submit your "Accept" recommendation.

Reviewer #1: All comments have been addressed

2. Is the manuscript technically sound, and do the data support the conclusions?

Reviewer #1: Yes

3. Has the statistical analysis been performed appropriately and rigorously? 

Reviewer #1: Yes

4. Have the authors made all data underlying the findings in their manuscript fully available?

Reviewer #1: Yes

5. Is the manuscript presented in an intelligible fashion and written in standard English?

Reviewer #1: Yes

6. Review Comments to the Author

Reviewer #1: (No Response)

7. PLOS authors have the option to publish the peer review history of their article (what does this mean?). If published, this will include your full peer review and any attached files.

Reviewer #1: No

---

## [Editor Report · Acceptance letter]

1 Apr 2022

PONE-D-21-06676R2 

Estimation of mean population salt intakes using spot urine samples and associations with body mass index, hypertension, raised blood sugar and hypercholesterolemia: Findings from STEPS Survey 2019, Nepal 

Dear Dr. Dhimal:

I'm pleased to inform you that your manuscript has been deemed suitable for publication in PLOS ONE. Congratulations! Your manuscript is now with our production department. 

Kind regards, 

on behalf of

Dr. PLOS Manuscript Reassignment 

Staff Editor

PLOS ONE